# Metagenomic Analysis Reveals the Mechanism for the Observed Increase in Antibacterial Activity of Penicillin against Uncultured Bacteria *Candidatus* Liberibacter asiaticus Relative to Oxytetracycline in Planta

**DOI:** 10.3390/antibiotics9120874

**Published:** 2020-12-05

**Authors:** Chuanyu Yang, Hanqing Hu, Yihong Wu, Xiongjie Lin, Goucheng Fan, Yongping Duan, Charles Powell, Veronica Ancona, Muqing Zhang

**Affiliations:** 1Institute of Fruit Tree Research, Fujian Academy of Agricultural Sciences, Fuzhou 350003, China; hanqinghu@126.com (H.H.); redone233@163.com (Y.W.); linxj_019@163.com (X.L.); guochengfan@126.com (G.F.); 2Citrus Center, Texas A&M University-Kingsville, 312 N. International Blvd, Weslaco, TX 785799, USA; Veronica.Ancona-Contreras@tamuk.edu; 3USHRL-USDA-ARS, 2011 S. Rock Rd, Fort Pierce, FL 34945, USA; yongping.duan@ars.usda.gov; 4IRREC-IFAS, University of Florida, 2199 S. Rock Rd, Fort Pierce, FL 34945, USA; capowell@ufl.edu; 5State Key Lab for Conservation and Utilization of Subtropical Agri-Biological Resources, Guangxi University, Nanning 530005, China

**Keywords:** citrus Huanglongbing, endophytic microbiome, oxytetracyline, penicillin, mode of action

## Abstract

Citrus huanglongbing (HLB) is a devastating disease for the citrus industry. The previous studies demonstrated that oxytetracycline and penicillin are effective antibiotics against *Candidatus* Liberibacter asiaticus (*C*Las). However, since *C*Las is uncultured, the mechanisms of action of antibiotics against *C*Las are still unclear. It was recently reported that the endophytic microbial communities are associated with the progression of citrus HLB after oxytetracycline and penicillin treatment. Therefore, we hypothesize that penicillin has greater antibacterial activity against *C*Las than oxytetracycline, which may be associated with the alteration of the structure and function of endophytic microbial communities in HLB-affected citrus in response to these antibiotics. To test this hypothesis, the microbiome of HLB-affected citrus leaves treated with these two antibiotics was analyzed using a metagenomic method. Our results indicate that the microbial structure and function in HLB-affected citrus were altered by these two antibiotics. The relative abundance of beneficial bacterial species, including *Streptomyces avermitilis* and *Bradyrhizobium,* was higher in penicillin-treated plants compared to those treated with oxytetracycline, and the relative abundance of the bacterial species (such as *Propionibacterium acnes* and *Synechocystis* sp PCC 6803) associated with *C*Las survival was lower for penicillin-treated plants compared to oxytetracycline-treated plants. These results indicate that penicillin has greater antibacterial activity against *C*Las. Based on the metagenomic analysis, this study elucidated the mechanism for the observed increase in antibacterial activity of penicillin against *C*Las. The data presented here are not only invaluable for developing eco-friendly and effective biocontrol strategies to combat citrus HLB, but also provide a method for revealing mechanism of antimicrobial against uncultured bacteria in host.

## 1. Introduction

Citrus huanglongbing (HLB) is one of the most devastating diseases of citrus. HLB spreads across most citrus-growing areas worldwide and has caused significant losses or declines in both production and profit of the citrus industry [1,2]. Three species of fastidious, phloem-restricted α-proteobacteria: “*Candidatus* Liberibacter asiaticus” (*C*Las), “*Candidatus* Liberibacter americanus”(*C*Lam), and “*Candidatus* Liberibacter africanus” (*C*Laf), which are transmitted by the psyllids *Diaphorina citri* or *Trioza erytreae*, are the major causative agents of HLB [1,2]. Of these three species, *C*Las is the most prevalent and the only species found in China and the United States. To date, there are no effective treatment strategies for infected trees in field, and there are no resistant commercial citrus varieties [3]. Chemotherapy has shown considerable promise for control of HLB in the short term.

The previous studies demonstrated that oxytetracycline and penicillin are effective antibiotics against *C*Las [4,5]. Oxytetracycline are short-acting antibiotics that inhibit bacterial growth by inhibiting translation. It passively diffuses through porin channels in the bacterial membrane, binds reversible to the bacterial 30S ribosomal subunit and prevents the aminoacyl tRNA from binding to the A site of the ribosome [6]. Oxytetracycline is only bacteriostatic, application of this antibiotic against *C*Las was later discontinued and requiring treatment needed to be repeated each year [7]. Penicillin as Beta-lactam antibiotic, inhibit the growth of sensitive bacteria by inactivating enzymes located in the bacterial cell membrane, known as penicillin binding proteins, which are involved in cell wall synthesis [8]. Penicillin are considered to be bactericidal [9]. HLB-affected citrus treated with penicillin in greenhouse, displayed no HLB-symptom and no *C*Las was detected in the plants [10]. However, since *C*Las is uncultured, the modes of action of penicillin and oxytetracycline against *C*Las are still unclear.

The plant microbiome in the rhizosphere, phyllosphere, and endosphere plays a key role in plant growth and health [11]. The structure and function of endophyte microbial communities in HLB-affected citrus are also altered by oxytetracycline and penicillin. The *C*Las strain Ishi-1 is reported not to be susceptible to oxytetracycline in vitro, but oxytetracycline treatment did eliminate a particular sub-community dominated by the families *Comamonadaceae*, *Flavobacteriaceae*, *Microbacteriaceae*, and *Pseudomonadaceae* to decrease *C*Las survival [12]. Several studies also indicated that when penicillin was applied to HLB-affected citrus by trunk injection, while the *C*Las titer was decreased, the abundance of *Actinomycetales*, *Frankiaceae*, and *Microbacteriaceae* was reduced and that of *Verrucomicrobia* and *Bacilli* was enhanced [13]. Therefore, the endophytic microbial communities were associated with the progression of citrus HLB following oxytetracycline and penicillin treatment.

We hypothesize that penicillin displayed greater antibacterial activity against *C*Las than oxytetracycline, which may be associated with the alteration of the structure and function of endophytic microbial communities in HLB-affected citrus in response to these antibiotics. Therefore, in this study, oxytetracycline or penicillin was applied to HLB-affected citrus by a foliar spray. Tissues from the treated trees were subjected to shotgun metagenomic analysis to reveal the mechanism of penicillin and oxytetracycline against *C*Las.

## 2. Materials and Methods

### 2.1. Plant Materials

Two-year-old healthy grapefruit (*Citrus paradisi*) seedlings were graft-inoculated with HLB-affected lemon (*C. limon*) scions and subsequently maintained in the greenhouse. After 10 months, typical HLB symptoms such as vein corking and blotchy mottles appeared on the leaves of the inoculated seedlings. HLB-affected citrus seedlings with typical HLB symptoms were then tested for the presence of *C*Las bacteria using quantitative real-time polymerase chain reaction (qPCR) with *C*Las-specific primers (HLBas, HLBr, and HLBp) [14].

### 2.2. Application of Antibiotics to HLB-Affected Citrus

Oxytetracycline hydrochloride (Oxy) (Aladdin Industrial Corporation, Shanghai, China) and penicillin G sodium (Pen) (Aladdin Industrial Corporation, Shanghai, China) were prepared as 500 mg/L chemical solution, respectively. Tap water was selected as control (CK). One liter of the prepared chemical solution (Oxy, Pen, and CK) was applied to 2 year-old HLB-affected grapefruit seedlings that exhibited typical HLB symptoms by foliar spray four times at 2-week intervals. Three biological replicates were performed for each treatment. The treated seedlings were grown at 28 °C ± 5 °C in an insect-proof greenhouse. Five leaf samples were collected from each treatment at 0 and 90 days after initial treatment (DAT), and DNA was extracted for qPCR and metagenomic analysis.

### 2.3. Graft-Based Assay

A graft-based assay was used to evaluate the effectiveness of avermectin, which is produced from *S. avermitilis*, against *C*Las. HLB-infected budsticks were collected from symptomatic rough lemons, and confirmed to be positive for *C*Las by real-time qPCR [4,14]. The budsticks (20 scions per treatment per concentration) were soaked in 500 mg/L avermectin solution overnight in a fume hood under ventilation and continuous fluorescent light. Each soaked budstick was cut into a 2-bud scion and grafted onto individual 2-year-old HLB-free grapefruit (*Citrus paradisi* “Duncan”) rootstock seedlings and the grafts were covered with plastic tape for three weeks. To enhance scion growth, new flush from the rootstocks was removed immediately after grafting. Grafted plants were shaded and maintained at 25 °C ± 2 °C in an insect-proof greenhouse. The effectiveness of the chemical treatments against *C*Las was determined by measuring the titer of *C*Las in both the grafted scion and the rootstock using qPCR. Briefly, five leaves were randomly sampled from the scion (rough lemon) and rootstock (grapefruit) 180 days after grafting. The leaves were washed in tap water and then rinsed three times with sterile water. Midribs were excised, frozen in liquid nitrogen, and stored at −80 °C for further processing. The midribs of five leaves from each sample were pooled and used for DNA extraction and subsequent qPCR analysis as described [4,14].

### 2.4. Genomic DNA Extraction and qPCR Analysis

Each leaf was rinsed three times with sterile water. Midribs were separated from the leaf samples and cut into pieces 1.0 to 2.0 mm in length. DNA was extracted from 0.1 g (fresh weight) of leaf midrib tissue using a DNeasy Plant Mini Kit (Qiagen, Valencia, CA, USA) according to the manufacturer’s protocol. qPCR was performed with primers and probes (HLBas, HLBr and HLBp) for *C*Las [14] using the ABI 7500 Fast Real-Time PCR System (Applied Biosystems, Foster City, CA, USA) in a 20 μL reaction volume containing 300 nM (each) target primer (HLBas and HLBr), 150 nM target probe (HLBp), and 1× TaqMan qPCR Mix (Applied Biosystems). The amplification protocol was 95 °C for 20 s followed by 40 cycles at 95 °C for 3 s and 60 °C for 30 s. All reactions were performed in triplicate and each run contained one negative (DNA from healthy plant) and one positive (DNA from a *C*Las-infected plant) control. The positive control was from the same sample, and was checked to ensure that the Ct remained constant. Data were analyzed using the ABI 7500 Fast Real-Time PCR System with SDS software.

### 2.5. DNA Extraction, Library Preparation, and Sequencing for Metagenomic Analysis

For shortgun metagenomic analysis, the leaf samples were chemically sterilized with ~5% sodium hypochlorite for 2 min to remove phyllosphere bacteria. DNA was extracted 90 days after the initial chemical treatment as described above. DNA from five leaves in each treatment was pooled in equal amounts from the three replicates. Afterward, the DNA was sheared into fragments of approximately 300 bp using an M220 Focused-ultrasonicator (Covaris Inc., Woburn, MA, USA) to build a paired-end library. DNA templates were then pretreated using a Tru-seq Kit according to the manufacturer’s instructions (https://www.illumina.com/). The libraries were pooled and loaded onto an Illumina cBot [15]. Pair-end sequencing (2 × 150 bp) was performed on a Hiseq2000 sequencer (Illumina) according to the standard protocol (https://www.illumina.com/).

### 2.6. Bioinformatics Analyses

Raw reads from metagenome sequencing were filtered, trimmed, and quality-controlled to generate clean reads, which were further trimmed using Sickle with the parameters -q 20 and -l 80 [16]. The trimmed reads were aligned to the sweet orange genome [17] using Bowtie [18] and the corresponding mapped reads were removed. Only reads that did not map to the citrus genome were retained for further analysis.

Metagenome taxonomy was assigned using small-subunit (SSU) rRNA gene tags and predicted proteins [19]. The SSU rRNA gene tags were obtained by aligning the raw reads with the Silva SSU rRNA database (http://www.arb-silva.de/) using an optimized E-value threshold of 1 × 10^−5^, an alignment length of 100 bp, and an identification cutoff of 80%. The taxonomic annotation of the samples was obtained by BLAST against the NCBI-nt database. For each protein-coding read, alignment with protein sequences in the NCBI-nr database was performed using the BLAST tool, and results with the best hit to the NCBI microbial taxonomy database were obtained.

The network of association between bacterial species within CK-, Oxy-, and Pen-treated samples was generated with SparCC methods [20]. The Shannon and Simpson biodiversity indices were calculated as:$$\mathrm{Shannon}\prime s\;\mathrm{index}\;=\;-\sum_{1}^{n}Fi\;LnFi\;\mathrm{and}\;\mathrm{Simpson}\prime s\;\mathrm{diversity}\;\mathrm{index}\;=\;1\;-\sum_{1}^{n}{Fi}^{2}$$
where n represents the richness or total number of families, *Fi* is the relative species abundance, *i*^th^ is the number of species was detected, and *Ln* is the natural logarithm.

Metagenes were predicted by MetaGeneMark. Non-redundant gene categories (uni-genes) were generated using CD-HIT-est with an identity cutoff of 95% [21]. To obtain taxonomic annotation for the unigenes, the protein sequences were aligned against the NCBI microbial NR database using DIAMOND software [22] with an E value cutoff of 1 × 10^−5^. Then, analysis of taxonomic and functional abundance was conducted by Kraken (0.10.5-beta) [23] with a custom database. Functional annotation was assigned to the unigenes by BLASTing against the NR, COG, KEGG, GO, Pfam, and Swissprot databases using DIAMOND software. The Venn diagram, heatmap and correlation analysis were performed using R packages (v3.2.0).

### 2.7. Statistical Analyses

Variance analysis was conducted to analyze the antibacterial activity of antibiotics. The data of antibiotics treatments were analyzed by Duncan’s multiple range test at *p*  <  0.05. All the data analysis was run in SAS software package (SAS V.9.1, SAS institute, Cary, NC, USA).

## 3. Results

### 3.1. Efficacy of Oxy and Pen against CLas

The antibiotics were applied to HLB-affected citrus as foliar sprays in greenhouse. Ninety days after the initial treatment, the *C*Las titer in the HLB-affected citrus was significantly reduced by both Oxy (*p* = 0.0007) and Pen (*p* = 0.0020) (Figure 1). For CK treatment, the *C*Las titer was not significantly changed (*p* = 0.1099) and remained at high levels (Figure 1). Pen (Ct value = 36.38 ± 2.33) treatment had significantly greater antibacterial activity against *C*Las compared to Oxy treatment (Ct value = 29.76 ± 1.45) (*p* = 0.0006).

### 3.2. Structure and Function of the Endophytic Microbial Community in HLB-Affected Citrus Leaves Treated with Oxy and Pen

Metagenomic analysis of leaf samples collected from HLB-affected citrus treated with Oxy, Pen, or CK yielded 2,254,884, 3,448,840, and 2,453,023 clean reads, respectively (Table 1). *Proteobacteria* was the most abundant phylum in all three treatments (Figure 2A). *Cyanobacteria* and *Actinobacteria* were abundant in plants treated with Oxy and Pen, respectively (Figure 2A). For CK-treated plants, *Candidatus* Liberibacter asiaticus species was more abundant than in Oxy- and Pen-treated plants. The relative abundance of *Candidatus* Liberibacter asiaticus following Oxy treatment was much higher than that seen for Pen treatment (Figure 2B). The relative abundance of *Streptomyces avermitilis* species was much higher after Oxy and Pen treatment, relative to that seen for CK (Figure 2B). In particular, the highest relative abundance of this species was seen in Pen-treated plants. In total, we identified 459 bacterial species, of which 151 were present in all three treatments (Figure 2C). CK had the highest number of species (327), while there were 226 and 208 species detected in Oxy- and Pen-treated plants, respectively (Appendix A). Moreover, Shannon’s and Simpson’s diversity indices both revealed that Oxy-treated plants had higher species diversity compared to Pen-treated plants (Figure 2D).

Under the antibiotics treatment, the correlation of *C*Las between the other 458 bacterial species was analyzed (Appendix A). The results indicated that 11 bacterial species, relative abundance of which was more than 0.0001, had significantly correlated with *C*Las (1 > ∣R∣ > 0.9, *p* < 0.05). The bacterial species including *Propionibacterium acnes*, *Pseudomonas putida, Synechocystis* sp_PCC_6803, *Sphingomonas wittichii*, *Staphylococcus epidermidis*, and *Deinococcus gobiensis*, had positively correlated with *C*Las (Figure 3). On the other hand, five bacterial species such as *Streptomyces avermitilis*, *Alteromonas macleodii*, *Bradyrhizobium japonicum*, *Bradyrhizobium* sp S23321, and *Bradyrhizobium oligotrophicum*, showed negative correlation with *C*Las (Figure 3).

Overall, 3380 unigenes were annotated against at least one of the public databases (Table 2). There were 1351 genes annotated against the COG database (Table 2). The relative abundance of energy production and conversion (C) and transcription (K) were enhanced by Oxy and Pen treatments (Figure 4A). In particular, the relative abundance of cell motility was lowest for Pen-treated plants.

In total, 1296 genes were annotated against the GO database (Table 2). The relative abundance of chloroplast, chloroplast thylakoid membrane, photosynthesis, nucleoside-triphosphatase activity, iron ion binding, 4 iron, 4 sulfur cluster binding, electron carrier activity, electron transport chain, photosynthetic electron transport chain, heme binding, respiratory electron transport chain, electron transporter, transferring electrons within cytochrome b6/f complex of photosystem II activity, and small ribosomal subunit were increased by both Oxy and Pen treatment (Figure 4B). In the Pen treatment, the relative abundance of membrane, NADH dehydrogenase (ubiquinone) activity, transport, thylakoid, DNA binding, DNA-directed RNA polymerase activity, plastid small ribosomal subunit, mitochondrial small ribosomal subunit, and transcription, DNA-templated were the highest relative to that for CK and Oxy (Figure 4B).

For KEGG annotation, 1796 genes were searched against the KEGG database (Table 2). The relative abundance of energy metabolism, nucleotide metabolism, and transcription were increased by both Oxy and Pen (Figure 4C). Notably, there were no unigenes detected for cell motility pathways in response to Pen.

### 3.3. Effect of Avermectin Streptomyces avermitilis on Candidatus Liberibacter asiaticus

*Streptomyces avermitilis* can produce avermectin. Therefore, the effectiveness of avermectin in reducing *C*Las titer was also evaluated by grafted-based assay. Although the *C*Las titer was not significantly different between avermectin- and tap water-treated (CK) scions (Figure 5), in rootstocks the *C*Las titer was much lower for the avermectin-treated plants compared to the CK plants (*p* = 0.0447).

## 4. Discussion

In this study, Pen displayed greater antibacterial activity against *C*Las than did Oxy (Figure 1). In addition, antibiotic treatment could change the microbial structure and function in HLB-affected citrus. Therefore, the alteration of *C*Las-associated microbiota may have contributed to the greater antibacterial activity of Pen.

Endophyte microorganisms can control plant pathogens [24] and enhance plant growth [25] through endophyte-mediated synthesis of natural compounds and metabolites. Our research indicated that *S. avermitilis* species were enriched in HLB-affected citrus following antibiotic treatment, especially relative abundance of this bacterial species was much higher in Pen treatment, relative to Oxy treatment (Figure 2B). Correlation analysis also indicated that this species may inhibit *C*Las growth (R = −0.92) (Figure 3). *Streptomyces. avermitilis* can produce avermectins [26], which have activity against helminths, insects, and arachnids [26,27]. Avermectins also have antibacterial activity against mycobacterial species [28]. The higher relative abundance induced by Pen treatment, may enhance avermectin production by *S. avermitilis*. In addition, the graft-based assay indicated that avermectins did not eliminate *C*Las, but did inhibit *C*Las transmission from scion to rootstock (Figure 5). Unigenes associated with cell motility were absent in Pen-treated tissues (Figure 4A), which may indicate that avermectin production by *S. avermitilis* induced by Pen treatment could inhibit mobility of endophytic bacteria, especially *C*Las. Therefore, the enrichment of *S. avermitilis* under Pen treatment, may enhance antibacterial activity against *C*Las (Figure 6) by inhibiting the mobility.

*Bradyrhizobium* sp. is a slow-growing nitrogen-fixing symbiotic bacterium of legumes and common root endophyte of other plants. This bacterium, which is considered to be a plant growth-promoting rhizobacterium (PGPR), is capable of colonizing the roots of non-legumes and produce phytohormones and siderophores as well as exhibit antagonistic effects toward several plant pathogens [29]. A recent study also demonstrated that *C*Las can decrease the relative abundance of most rhizoplane-enriched genera, such as *Bradyrhizobium*, and reduce the relative abundance of the functional attributes involved in microbe-plant interactions [30]. Therefore, *Bradyrhizobium* is beneficial for citrus health. In this study, the relative abundance of three *Bradyrhizobium* species was highest in the Pen treatment group compared with Oxy and CK treatment groups (Figure 6). Moreover, a previous study indicated that unigenes in *Bradyrhizobium* in healthy citrus rhizoplane-upregulated was enriched in the categories of “transcription” and “metabolism|unclassified” compared to HLB-affected citrus [30]. Our study also showed that the relative abundance of several GO terms, including DNA binding, DNA-directed RNA polymerase activity and transcription, and DNA-templated, which were associated with transcription, were enhanced by the antibiotics and the relative abundance in the Pen treatment group was much higher than in the Oxy treatment group (Figure 4B). Therefore, enhancement of the relative abundance of transcription by enriched *Bradyrhizobium* may contribute to the greater antibacterial activity of Pen against *C*Las (Figure 6).

Results from an earlier study suggested that *C*Las could use ecological services derived from *C*Las-associated microbiota to colonize the host and generate pathogen-associated community that stimulates disease development [12]. Our study found that several bacterial species including *Propionibacterium acnes*, *Pseudomonas putida*, *Synechocystis* sp PCC 6803, and *Staphylococcus epidermidis*, have a positive association with *C*Las. The relative abundance of these bacteria was decreased by both antibiotics, and the relative abundance were much lower in Pen treatment, relative to that in Oxy treatment (Figure 3).

*Propionibacterium acnes* as Gram-positive bacterium, belonged to Actinobacteria. In previous study, although *C*Las was not isolated in axenic culture, it was isolated in co-cultures with actinobacteria closely related to *P. acnes*. The co-cultures remained after attempts to purify the cultures by single-colony isolation, suggesting that the bacteria might be mutually beneficial to each other in culture [31]. In addition, the Znu system, a member of ABC transporter family, is critical for survival and pathogenesis of *C*Las [32]. Previous finding indicated a role in zinc uptake for the ZnuA, ZnuB, and ZnuC proteins from *Sinorhizobium meliloti* as well as one homologous gene cluster in *C*Las, suggesting this system was associated with growth and pathogenesis of *C*Las [33]. The sequence alignment of *C*Las-ZnuA2 sequence showed 24% identity with *Synechocystis* [34]. In this paper, the bacterial species *Synechocystis* sp PCC 6803 may share functional znuA homologues that encode for a high affinity zinc uptake system associated with growth and virulence of *C*Las. Therefore, the lower relative abundance of bacteria species associated with *C*Las survival following Pen treatment may have resulted in greater antibacterial activity against *C*Las.

However, it was found that *Pseudomonas putida* could significantly reduce the population of viable *C*Las in HLB symptomatic leaves [35]. In this study, relative abundance of *C*Las in HLB-affected citrus leaves was decreased by the antibiotics, as well as that of *Pseudomonas putida* (Appendix A). Meanwhile, there are still no reports about the function of *Staphylococcus epidermidis* in citrus. Therefore, whether *Pseudomonas putida* and *Staphylococcus epidermidis* are associated with *C*Las survival is still unknown. It will be studied in further research.

## 5. Conclusions

Although it is difficult to elucidate the mechanism of action of an antibiotic against uncultured bacteria *C*Las, metagenomic analysis can provide new insight into the endophytic microbial community in HLB-affected citrus following antibiotic treatment, which was associated with antibacterial activity of the antibiotic against *C*Las. Our study indicated that the relative abundance of beneficial bacterial species, including *S. avermitilis* and *Bradyrhizobium,* was higher in Pen-treated plants compared to those treated with Oxy, and the abundance of the bacterial species (such as *Propionibacterium acnes* and *Synechocystis* sp PCC 6803) associated with *C*Las survival was lower for Pen-treated plants compared to Oxy-treated plants, indicating that Pen has greater antibacterial activity against *C*Las. Moreover, the beneficial bacteria in this study will be a potential candidate for biocontrol of citrus HLB. Now, isolation and identification of these two beneficial bacteria are undergoing. *Propionibacterium acnes* and *Synechocystis* sp PCC 6803 can be selected for target bacteria to decrease *C*Las survival. Therefore, this study not only developed a novel strategy for studying modes of action of antimicrobials against uncultured bacteria, but also provided valuable insight for developing eco-friendly and effective strategies to combat citrus HLB or other plant bacteria pathogen.

## Figures and Tables

**Figure 1 antibiotics-09-00874-f001:**
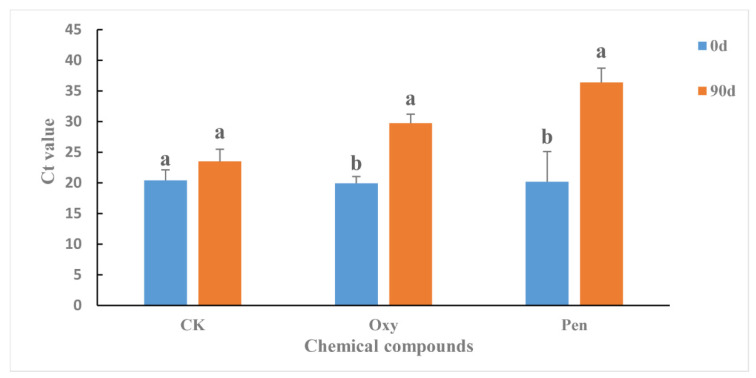
Ct value in HLB-affected citrus treated with oxytetracycline (Oxy) and penicillin (Pen). Different letters represented significantly differences at the level of 0.05 (*p* ≤ 0.05).

**Figure 2 antibiotics-09-00874-f002:**
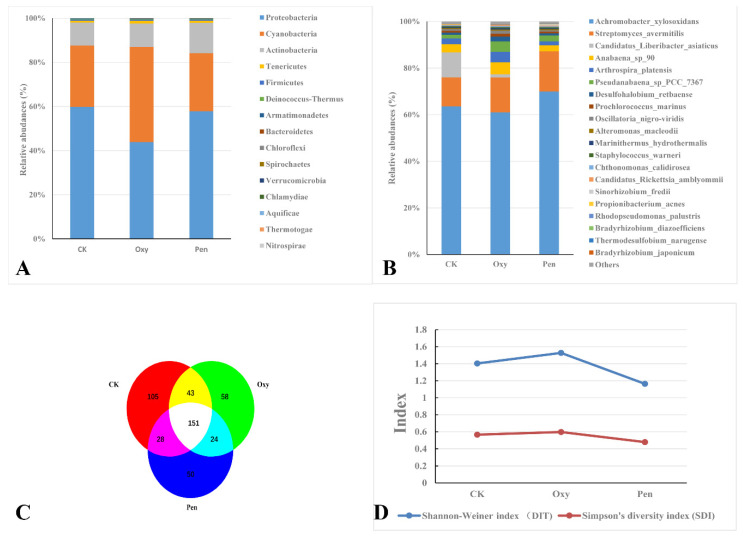
Bacterial species present after antibiotic treatments of HLB-affected citrus plants. Taxonomic abundance of microbial community at the (**A**) phyla and (**B**) species rank in HLB-affected citrus treated with oxytetracycline (Oxy) or penicillin (Pen). (**C**) Venn diagram representing bacterial species in the endophytic microbial community in HLB-affected citrus treated with tap water (CK), oxytetracycline (Oxy), or penicillin (Pen). (**D**). Simpson’s diversity index (SDI) and Shannon–Weiner index (DIT) of the endophytic microbial community in HLB-affected citrus treated with tap water (CK), oxytetracycline (Oxy), or penicillin (Pen).

**Figure 3 antibiotics-09-00874-f003:**
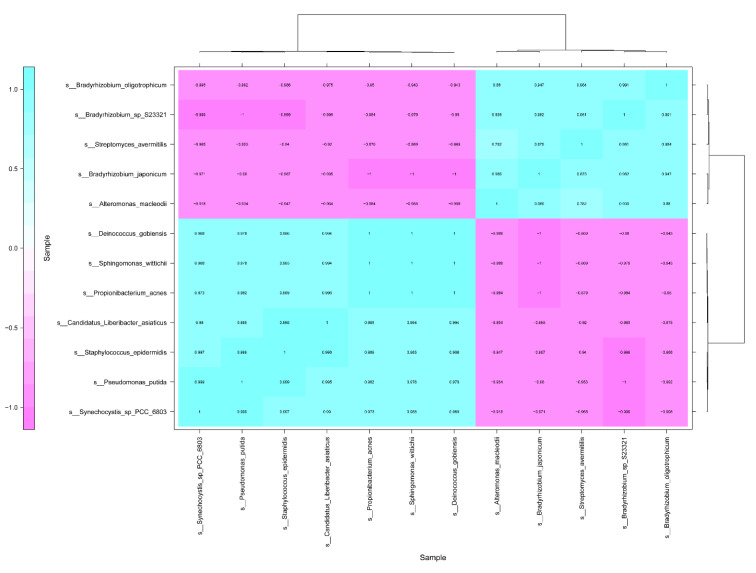
Correlation analysis of *Candidatus* Liberibacter asiaticus between other bacterial species in HLB-affected citrus leaves in response to the antibiotics.

**Figure 4 antibiotics-09-00874-f004:**
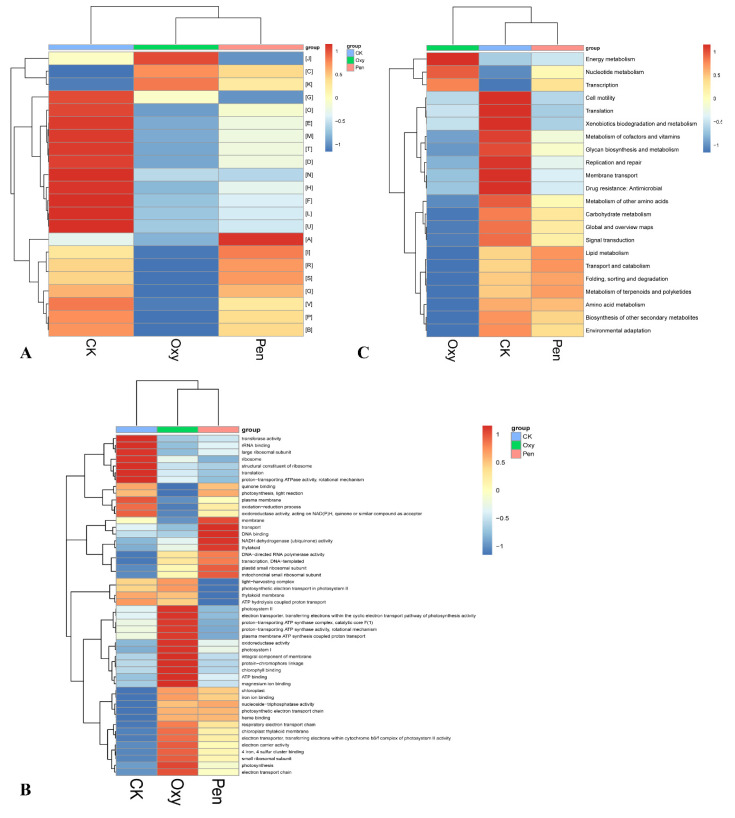
Heatmap of COG **(A)**, GO **(B)**, and KEGG **(C)** function comparison of the microbiome in HLB-affected citrus in response to oxytetracycline (Oxy) and penicillin (Pen) treatment.

**Figure 5 antibiotics-09-00874-f005:**
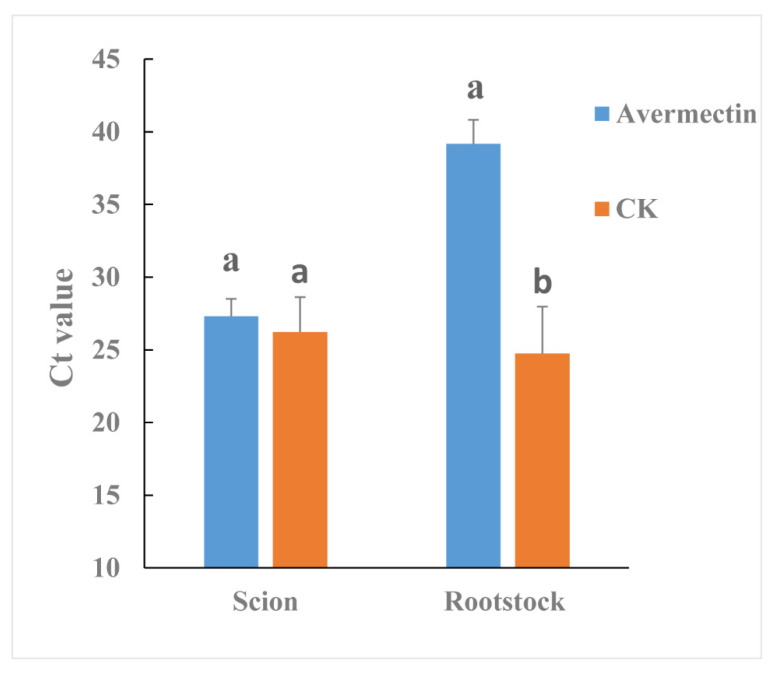
Effectiveness of avermectins against *C*Las as evaluated by a grafted-based assay. Different letters represented significantly differences at the level of 0.05 (*p ≤* 0.05).

**Figure 6 antibiotics-09-00874-f006:**
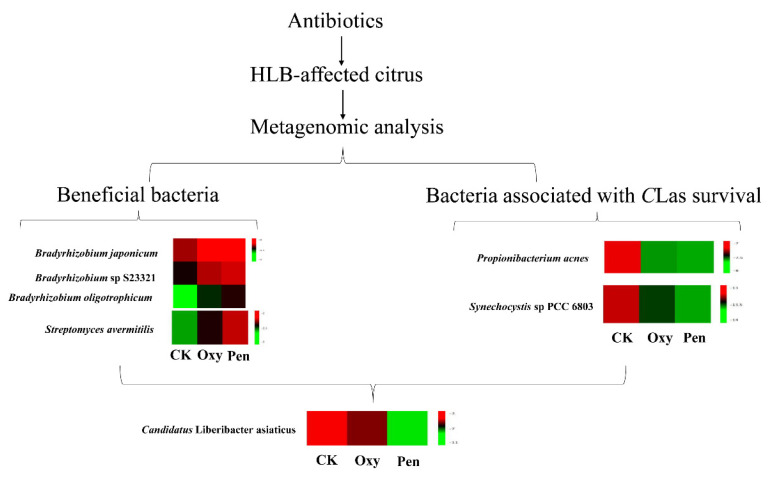
The mechanism of the antibiotics (oxytetracycline (Oxy) and penicillin (Pen)) against *Candidatus* Liberibacter asiaticus (*C*Las) based on metagenomic analysis.

**Table 1 antibiotics-09-00874-t001:** Summary of metagenome data.

Treatment	Raw Data (bp)	Clean Data (bp)	Clean Reads	GC (%)	Q20 (%)	Q30 (%)
CK	794,635,500	676,465,200	2,254,884	42.81	93.88	87.72
Oxy	1,228,026,300	1,034,652,000	3,448,840	42.13	94.84	89.12
Pen	853,398,900	735,906,900	2,453,023	40.74	94.39	88.5

**Table 2 antibiotics-09-00874-t002:** Number of unigenes annotated against COG, GO, KEGG, Nt, Pfam, and Swissprot databases.

Anno_Database	Annotated_Number	300 ≤ Length < 1000	Length ≥ 1000
COG_Annotated	1351	146	2
GO_Annotated	1296	133	1
KEGG_Annotated	1796	147	2
Nt_Annotated	2145	216	5
Pfam_Annotated	2222	231	4
Swissprot_Annotated	2548	223	4
All_Annotated	3380	242	5

## Data Availability

The data that support the findings of this study are available from the corresponding author upon reasonable request.

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
