# Peer review of "Metagenomic Analysis Reveals the Mechanism for the Observed Increase in Antibacterial Activity of Penicillin against Uncultured Bacteria Candidatus Liberibacter asiaticus Relative to Oxytetracycline in Planta"

_antibiotics, 2020, doi:10.3390/antibiotics9120874_

Round 1

Reviewer 1 Report

The MS titled: Metagenomic Analysis Reveals the Mechanism for the Observed Increase in Antibacterial Activity of Penicillin against Uncultured Bacteria Candidatus Liberibacter asiaticus Relative to Oxytetracycline in Planta

By: Chuanyu Yang, Hanqing Hu, Yihong Wu, Xiongjie Lin, Goucheng Fan, Yongping Duan, Charles Powell, Veronica Ancona, and Muqing Zhang

This MS is examining the community and function of endophytic bacteria, inhabiting citrus trees infected by Liberibacter, as effected by different antibiotics.

The MS is well written and structured and it seems that the authors done a lot of work in both greenhouse experiments and molecular/ bioinformatic analyses.

However, a very important control of uninfected trees with and without antibiotics is missing. Without this control, the hypothesis that the changes of endophytes communities and function is connected to the disease and not to the application of antibiotics can not be tested.

Therefore, I sadly recommend rejecting this paper

Author Response

This MS is examining the community and function of endophytic bacteria, inhabiting citrus trees infected by Liberibacter, as effected by different antibiotics.

The MS is well written and structured and it seems that the authors done a lot of work in both greenhouse experiments and molecular/ bioinformatic analyses.

However, a very important control of uninfected trees with and without antibiotics is missing. Without this control, the hypothesis that the changes of endophytes communities and function is connected to the disease and not to the application of antibiotics can not be tested.

Therefore, I sadly recommend rejecting this paper

Response: Thank you for your reviewing this manuscript. This study focus on revealing mode of action of antibiotics against uncultured bacteria Candidatus Liberibacter asiaticus (CLas) based on metagenomic analysis. Therefore, we just selected HLB-affected citrus plants as materials, and we can understand how antibiotics affect CLas via alteration of bacterial community and structure in HLB-affected citrus. if we want to decipher microbiome of HLB-affected citrus in response to antibiotics, the control of uninfected citrus is very important. Therefore, Different research purpose maybe need different control.

Reviewer 2 Report

This manuscript assesses the effect of two antibiotics on the uncultured bacteria Candidatus Liberibacter asiaticus (CLas), one of the major causative agents of the Citrus huanglongbing (HLB) disease. The effect of those antibiotics was also evaluated on the microbial community in HLB-affected citrus leaves to gain further insight into the mode of action of those antibiotics. Data from this study revealed that penicillin has greater antibacterial activity against CLas compared to Oxytetracycline. Using metagenomic analysis authors have provided interesting data showing that the effect of those two antibiotics on CLas is potentially associated with their effect on other beneficial bacteria present within the citrus microbiome.

Overall, the manuscript is very well organized and written. It contains interesting findings that might be of great interest to Antibiotics readers, particularly research groups working on citrus HLB or other bacterial plant pathogens. Some minor revisions are requested before acceptance for publication in Antibiotics. Here are some specific edits that should be done to improve the current version of the manuscript:

Minor revisions:

-Line 23: Precise what does the abbreviation” CLas” stands for in the abstract.

-Line 48: Add an “s” to “spread”.

-Line 76: You should precise that the abbreviation “Pen” refers to ‘’Penicillin”

-Line 80 and 81: Authors should be consistent when they are using abbreviations. They should either use the abbreviation “Pen” in the whole manuscript or continue using the unabbreviated word.

-Line 99: Replace “1L” with “One liter”. Never start a sentence with a number and an abbreviated unit.

- In materials and methods, the authors should add a paragraph to explain how statistical analyses were conducted and the software used for those analyses.

-In Figure 1, the authors are showing the same set of data in two different graphs to show the statistical differences between treatments. I suggest just keep one of the graphs and find a different and easier way to show statistical differences.

- Line 284: “S. avermitilis …arachnids”: Do not start a sentence with an abbreviation, replace “S.” with “Streptomyces”.

- Line 314: Replace “bacteria” with “bacterial”.

- Line 352: I believe there is a typo, or a word missed before “And”, please modify.

- The conclusion paragraph (Lines 343-356) summarizes the findings of this study, however, the reviewer would want to see a stronger concluding paragraph that highlights the importance of this study in the field and states some future directions and potential applications on other bacterial plant pathogens.

Author Response

Line 23: Precise what does the abbreviation” CLas” stands for in the abstract.

Response: thank you for your comments, we have revised it.

-Line 48: Add an “s” to “spread”.

Response: thank you for your comments, we have revised it.

-Line 76: You should precise that the abbreviation “Pen” refers to ‘’Penicillin”

Response: thank you for your comments, we have precised abbreviation “Pen” refers to”Penicillin”

-Line 80 and 81: Authors should be consistent when they are using abbreviations. They should either use the abbreviation “Pen” in the whole manuscript or continue using the unabbreviated word.

Response: thank you for your comments, , we have revised it and used Penicillin in introduction section. However, Oxytetracycline hydrochloride (Oxy) and penicillin G sodium (Pen) were applied in our experiment, the abbreviations were used in materials and method, results and discussion sections.

-Line 99: Replace “1L” with “One liter”. Never start a sentence with a number and an abbreviated unit.

Response: thank you for your comments, we have revised it.

- In materials and methods, the authors should add a paragraph to explain how statistical analyses were conducted and the software used for those analyses.

Response: thank you for your comments, we have added a paragraph about statistical analyses in Materials and Methods section.

-In Figure 1, the authors are showing the same set of data in two different graphs to show the statistical differences between treatments. I suggest just keep one of the graphs and find a different and easier way to show statistical differences.

Response: thank you for your comments, we have revised the Figure 1 and results section in updated manuscript.

- Line 284: “S. avermitilis …arachnids”: Do not start a sentence with an abbreviation, replace “S.” with “Streptomyces”.

Response: thank you for your comments, we have revised it.

- Line 314: Replace “bacteria” with “bacterial”.

Reponse: thank you for your comments, we have replaced “bacteria” with “bacterial”

- Line 352: I believe there is a typo, or a word missed before “And”, please modify.

Response: thank you for your comments, we have modified it.

- The conclusion paragraph (Lines 343-356) summarizes the findings of this study, however, the reviewer would want to see a stronger concluding paragraph that highlights the importance of this study in the field and states some future directions and potential applications on other bacterial plant pathogens.

Response: thank you for your comments, we have revised it.

Reviewer 3 Report

In this manuscript, the authors performed the metagenomic analysis to try to demonstrate the mechanism of antibacterial activity against CLas by penicillin. The topic and content should be of interest to many readers. However, only metagenomic analysis is not enough to prove the relationship between the bacterial species and antibacterial activity against CLas by penicillin. Additionally, the authors need to repeat the treatment to further verify the enhancement of these bacterial species.

I mainly have two questions:

How the authors determine these bacterial species in the penicillin-treated tissues to correlate with the antibacterial activity against CLas by penicillin? Maybe penicillin just changed the abundance of these bacterial species, but abundance change of these bacterial species doesn’t play important role in antibacterial activity against CLas. The authors performed the Graft-based assay to try to demonstrate the role of avermectin in antibacterial activity, but that is not enough to prove the role of S. avermitilis in penicillin-treated tissues.  

Additionally, I want to know whether these enhanced bacterial species always exist and are enhanced in penicillin-treated tissues. Did the authors repeat the treatment (I mean not one batch of treatment) and at least use PCR to detect the enhancement? Maybe the penicillin-treatments enhance other bacterial species in another treatment?

Author Response

  1. How the authors determine these bacterial species in the penicillin-treated tissues to correlate with the antibacterial activity against CLas by penicillin? Maybe penicillin just changed the abundance of these bacterial species, but abundance change of these bacterial species doesn’t play important role in antibacterial activity against CLas. The authors performed the Graft-based assay to try to demonstrate the role of avermectin in antibacterial activity, but that is not enough to prove the role of S. avermitilis in penicillin-treated tissues.

Response: thanks for your question. Based on Correlation analysis of CLas between other bacterial species in HLB-affected citrus leaves in response to the antibiotics this study indicated that the relative abundance of beneficial bacterial species, including S. avermitilis and Bradyrhizobium, was higher in Pen-treated plants compared to those treated with Oxy. In our future study, we are isolating these two beneficial bacteria for identifiying the antibacterial activity against CLas.

  1. Additionally, I want to know whether these enhanced bacterial species always exist and are enhanced in penicillin-treated tissues. Did the authors repeat the treatment (I mean not one batch of treatment) and at least use PCR to detect the enhancement? Maybe the penicillin-treatments enhance other bacterial species in another treatment?

Response: thanks for your questions, we also do the research about microbiome of HLB-affected citrus in response to penicillin via PhyloChip G3. The results also indicated the abundance of Streptomyces and Bradyrhizobium were enhanced by Penicillin. These data will be prepared for the other paper. Therefore, we think these two bacteria may be a potential candidates for biocontrol of citrus HLB.

Round 2

Reviewer 3 Report

Although the authors said: "we also do the research about microbiome of HLB-affected citrus in response to penicillin via PhyloChip G3. The results also indicated the abundance of Streptomyces and Bradyrhizobium were enhanced by Penicillin. These data will be prepared for the other paper. "

But I still think the authors should present some repeated data about the enhanced abundance of bacterial species in penicillin-treated tissues in this manuscript.  

This manuscript is a resubmission of an earlier submission. The following is a list of the peer review reports and author responses from that submission.